# The Temperature-Associated Effects of Rift Valley Fever Virus Infections in Mosquitoes and Climate-Driven Epidemics: A Review

**DOI:** 10.3390/v17020217

**Published:** 2025-02-01

**Authors:** Faustus A. Azerigyik, Shelby M. Cagle, William C. Wilson, Dana N. Mitzel, Rebekah C. Kading

**Affiliations:** 1Center for Vector-Borne Infectious Diseases, Department of Microbiology, Immunology & Pathology, Colorado State University, Fort Collins, CO 80523, USA; faustus.azerigyik@colostate.edu (F.A.A.); shelby.cagle@colostate.edu (S.M.C.); 2Foreign Arthropod-Borne Animal Diseases Research Unit, Agricultural Research Service, United States Department of Agriculture, 1515 College Ave., Manhattan, KS 66502, USA; william.wilson2@usda.gov (W.C.W.); dana.mitzel@usda.gov (D.N.M.)

**Keywords:** temperature effects, *Phlebovirus*, vector competence, *Culex* and *Aedes* species, mosquito genetics and immunity, *Orthobunyavirus*, vector-borne diseases, transovarial transmission, reassortment

## Abstract

Rift Valley fever virus (RVFV) is a mosquito-borne zoonotic disease within the genus *Phlebovirus*. Symptoms of the disease in animals range from moderate to severe febrile illness, which significantly impacts the livestock industry and causes severe health complications in humans. Similar to bunyaviruses in the genus *Orthobunyavirus* transmitted by mosquitoes, RVFV progression is dependent on the susceptibility of the physical, cellular, microbial, and immune response barriers of the vectors. These barriers, shaped by the genetic makeup of the mosquito species and the surrounding environmental temperature, exert strong selective pressure on the virus, affecting its replication, evolution, and spread. The changing climate coupled with the aforementioned bottlenecks are significant drivers of RVF epidemics and expansion into previously nonendemic areas. Despite the link between microclimatic changes and RVF outbreaks, there is still a dearth of knowledge on how these temperature effects impact RVF transmission and vector competence and virus persistence during interepidemic years. This intricate interdependence between the virus, larval habitat temperatures, and vector competence necessitates increased efforts in addressing RVFV disease burden. This review highlights recent advancements made in response to shifting demographics, weather patterns, and conveyance of RVFV. Additionally, ongoing studies related to temperature-sensitive variations in RVFV–vector interactions and knowledge gaps are discussed.

## 1. Introduction

Rift Valley fever virus (RVFV) is a vector-borne virus within the *Phenuiviridae* family in the order *Bunyavirales* that causes disease in both humans and animals. Primarily, outbreaks of RVF disease cause febrile illness, stillbirth in breeding animals, bloody diarrhea, and death, which have significant economic impacts [1]. Although human infections are generally self-limiting illnesses (estimated to be <1%) [2], infection can result in febrile episodes with protracted convalescence, severe hemorrhagic fever, liver damage, and death [3,4]. Incidences of RVF involving both humans and animals were documented in 32 countries between 2000 and 2019. Official reporting sources linked RVFV to 72,960 animal cases and 17,810 animal fatalities, as well as 5228 suspected human cases resulting in 987 deaths during this period [5]. Because of the potential for serious harm to human and animal health, the National Institutes of Health (NIH) lists RVFV as a Category A Priority Pathogen, further highlighting its relevance in biodefense research [6]. It is an overlap select agent with the Centers for Disease Control (CDC) and the United States Department of Agriculture (USDA) and is also included on the WHO Blueprint of Priority Diseases due to its potential for global dissemination [7,8]. In an effort to prepare for the control of epizootics caused by the virus, vaccine programs using live-attenuated and inactivated forms of RVFV as an alternative to manage disease burden have been ongoing [9,10]. Veterinary vaccines have been approved for use in some endemic countries including Rwanda, Kenya, and Tanzania [9,10,11]. These include inactivated RVFV vaccine strains, Smithburn, and Clone-13 [11]. These vaccines are often only used when risk models predict ideal conditions for RVF transmission [12]. In the meantime, the unpredictability of outbreaks and the practicality of rolling out comprehensive and economically viable immunization strategies [13] are hindered by inactivated vaccine manufacturing, which requires high containment facilities and numerous primary and booster doses to establish protective immunity [1]. In contribution to this drawback, these vaccines are not yet authorized for use in many endemic regions [9]; there remains poor attenuation in immunocompromised animals; and residual virulence in pregnant and neonatal ruminants has contributed to this drawback [14,15,16].

In addition to vaccination, mitigation strategies must incorporate methods to target mosquitoes that transmit RVF. Cohort studies that have implicated previously suspected species [17,18,19] and experimental data [19,20,21,22,23,24,25,26,27,28] show that mosquito species within the *Culex*, *Aedes*, *Mansonia*, and *Anopheles* genera are susceptible to RVFV infection with the ability to transmit the virus via their blood-feeding activities [29,30]. There has been an increased frequency of eco-climatic changes such as temperature, rainfall, humidity, etc., driving the growing danger of the spread of mosquito-borne RVFV transmission among mosquito species [31,32,33,34,35,36,37,38], highlighting the need for more studies to determine the extent to which these changes potentially alter vector competence and viral transmission. Generally, these climatic events impact the survivorship of many arthropod vectors by altering larval habitat conditions and the vector competence of mosquito species [39].

RVFV outbreaks have been driven by or linked to increasing global climate changes and anthropogenic activities (farming and expansion of settlements) [34]. Additionally, uncertainty associated with these rapidly changing conditions means that vectors may be exposed to variable stimuli or stressors in their natural habitats, affecting the consistency of results and interpretations of vector study outcomes [40]. Understanding how the virus transmission dynamics are potentially altered by changes in temperature is a significant consideration for forecasting future disease outbreaks and incursions into nonendemic regions [41]. Despite its global threat, the control of RVFV is increasingly ominous due to a lack of adequate understanding of the effects of microclimatic changes (the range of environmental variables recorded in specific locations) in environmental temperatures and its impact on mosquito vector competence and disease spread. This invariably adds to the daunting prospect of having to implement suitable model(s) that accurately predict future epidemics. Accordingly, this review highlights major advances with respect to the following: (i) the virus evolution, transmission cycle, and virus maintenance in nature (ii) changing thermal conditions and their impact on virus–vector interactions, and finally (iii) proposed new directions for future studies that potentially improve our understanding of climate-driven transmission and vector competence of RVFV, especially in temperate regions (Figure 1).

## 2. RVFV Zoonosis and the Role of Mosquitoes in Driving the Threat of Disease

The movement of people, animals, and freight presents opportunities for vector distribution and RVFV introduction into new regions [42]. Retrospective surveillance examining the direct link between RVFV-endemic nations and nonendemic areas like the USA revealed that infected travelers via flights and the unintended shipping of infected adult mosquitoes present the most significant risks of introducing RVFV into the USA [43]. The distribution of RVFV among mosquitoes, ungulates, and humans during interepizootic periods is discontinuous at several scales due to regional variations in environmental support for the development of numerous mosquito vectors [44]. Additionally, experimental data demonstrate a widespread and ability of native mosquito species across multiple genera in areas of potential invasion to be capable of RVFV transmission [18,20,21,22,45]. This is highlighted by reports that many North American species of mosquito have been demonstrated in the laboratory to be competent in the transmission of RVFV [22,46,47,48]. Mosquitoes capable of transmitting the virus from infected individuals risk disease spread among human and animal populations [49]. There have been documented cases of returning travelers infected with RVF, highlighting the risks for introduction to naive areas, leading to the risk of transmission by native competent vectors [50,51]. The spread of the virus is expected to have a highly complicated ecology between regions due to variable climate patterns, further presenting a window of opportunity to investigate the effect of virus and vector genetics on successive incursions in native mosquito populations.

The invasion of highly competent mosquito species into new areas also poses a risk for virus establishment and transmission, particularly if those species are tolerant of temperate climate conditions. For instance, with recent global expansion, *Aedes j. japonicus* and its sibling species, *Ae. koreicus* [52], have been determined to have infection and dissemination rates of over 90% and 84%, respectively; for mosquitoes fed on viremic hamsters, *Ae. j. japonicus* was found to be particularly effective in the transmission of RVFV [52]. *Aedes japonicus* is widely distributed in the Eastern United States, with large areas of suitable habitat for potential expansion [53,54]. *Ae. japonicus* is also adapted to harsh winters and snow and its larvae have been discovered in 4 °C water [55]. While it is particularly crucial to investigate the significance of temperate species in contributing to RVFV overwintering, it remains unclear whether RVFV is able to overwinter successfully in various genera of temperate mosquitoes, as *Culex* species have been observed to overwinter as adults, while *Ae. triseriatus* overwinters in prepupal stages [56,57]. The discovery of *Ae. triseriatus* (a tree-hole ovipositing mosquito) larvae provided the first indication that transovarial transmission (TOT) serves as an overwintering strategy for another bunyavirus, LaCrosse virus (LACV) [58,59]. Among the bunyaviruses, TOT appears to be a commonly utilized overwintering mechanism, albeit not the only one, with cases of viral-induced overwintering (TOT) occurring in temperate regions where winter bottlenecks can interrupt continuous horizontal maintenance, contributing to an evolving overwintering strategy [60] (Appendix A). While climate and temperature in North America and other potential introductory hotspots may likely differ significantly from those in endemic regions of Africa due to colder climates during winter periods, it is vital to consider these contextual factors when interpreting the threat of RVFV prevalence in novel composite ecotypes between temperate and tropical regions. For example, data-driven theoretical assessment of the receptivity for RVFV suggests that, although mosquito vectors are poorly propagated during winter months, the persistence of RVFV through these periods is hypothesized to rely on mechanisms that facilitate long-term maintenance between epidemics, such as TOT via mosquitoes restricted conceptually to areas with early increases in the vector population after the winter months [61].

### 2.1. Transmission Mechanisms and RVFV Persistence—A Leverage Trail During Interepidemic Periods

RVFV persistence during interepidemic periods (IEPs) via low levels of horizontal transmission has been shown in sero-epidemiological surveillance studies [62,63,64]. This has been demonstrated in previous studies in animal and human populations throughout Africa [65], in which seroconversion was documented in the absence of an outbreak. Sheep and goats in East Africa had RVFV incidences of 3–18% between 1998 and 2005 during an IEP, characterized by arid weather conditions with very low vector densities and low or undetectable levels of disease activity [66], until the resurgence of favorable conditions for vector transmission. In Zambia, yearly seroconversions were detected in a sentinel cattle herd [62]; in Tanzania, IgM antibodies against RVFV were detected in humans 6–7 years after an outbreak [64]; and in Senegal, virus isolations were made from mosquitoes and seroconversions of sentinel sheep were detected during an interepidemic period [63]. In these regions of endemicity, RVFV appears to be able to persist, with transmission occurring through an interepidemic period or even yearly frequency.

Vertical transmission in mosquitoes is reasoned to be a potential route for the virus to persist endemically in some areas, even through inhospitable environmental conditions [67,68,69]. Although many arboviruses are transmitted via a horizontal mechanism, cycling between the blooding-sucking arthropods and vertebrate hosts, vertical transmission is prevalent among viruses within the *Bunyavirales* order [60] (Appendix A). Transovarian or trans-stadial transmission of arthropod-borne viruses is implicated to be quite frequent and an evolutionary mechanism as a means of survival and persistence during periods of subsisting or severe weather conditions (e.g., overwintering) [60,70]. Against this backdrop, it has been hypothesized that orthobunyaviruses may have higher vertical transmission rates because they are vertically transmitted via transovarial transmission. Similar to other genera within the order *Bunyavirales*, well-known chronic viral infections such as LaCrosse virus (*Orthobunyavirus* genus), Crimean–Congo hemorrhagic fever virus (*Orthonairovirus* genus), and Sandfly fever (Toscana virus: *Phlebovirus* genus), demonstrate increasing evidence of a cycle of transovarial and trans-stadial transmission, often occurring among the arthropod vectors in ticks (*Orthonairovirus* group) [71], sandflies, or mosquitoes (*Phlebovirus*) [72]. Phleboviruses that were found many years ago but were previously believed to be insignificant human pathogens are now being reevaluated in light of the discoveries of a novel phlebovirus that causes severe fever with thrombocytopenia syndrome (SFTS virus, SFTSV) [73] and Heartland virus (HRTV) [74], which can be transferred trans-stadially in infected nymphal ticks to uninfected larvae through co-feeding [75,76]. Also, some aseptic meningitis cases in central Italy that went undetected for several years were later determined to be caused by Toscana virus (TOSV), transmitted by phlebotomine sandflies and capable of both horizontal and transovarial transmission [77]. These findings show that TOT is well documented in this order of viruses, and consequently, TOT may be a strategy important in RVFV maintenance as well [22,60]. The ability of these viruses to maintain transovarial transmission in their arthropod hosts may be considered an immune evasion mechanism to which immune sensitivities favor virus persistence in the host(s) [70,78]. This especially occurs during periods of low transmission characterized by low-level viral titers or enzootic activity [70]. Furthermore, species within the *Culex* genus are exemplified by their ability to adapt to new environments and a propensity to deposit eggs in shallow crevices in geomorphic surfaces [79,80]; this is characteristic of *Aedes* species as well. The eggs of *Aedes* spp. are drought-resistant, allowing the virus to persist in arid soils for extended periods until such a time when the eggs hatch after intense periods of rain [79]. Following periods of heavy rain and flooding during El Niño climatic seasons, the emergence of large numbers of *Aedes* (primary vectors) followed by *Culex* mosquitoes (secondary vectors) precipitate increases in RVFV transmission [81]. In the context of laboratory settings, *Ae. melanimon* and *Ae. vexans* have been shown to have a considerably lower likelihood of ovarian infections with RVFV, relative to *Cx. tarsalis* [26]. However, compared to *Culex* eggs, *Aedes* eggs are often more resistant to desiccation, which might provide vertically transmitted viruses a selective advantage [82]. Other reports indicate that in arid climates, as opposed to equatorial or mild temperate climates, *Aedes* mosquitoes exhibit more efficient vertical transmission rates [83]. Assessment of vertical transmission efficiency in RVFV vectors, particularly with regard to environmental influences and desiccation, is an area of research that needs further investigation.

## 3. Diurnal Temperature Changes as a Frontier in RVFV Spatial Reach

### 3.1. Daily Environmental Temperatures Alter RVFV Transmission Dynamics

#### 3.1.1. Theoretical Findings

The unpredictability of weather patterns, the susceptibility of mosquito species to RVFV across different genera (Appendix A), and their ability to colonize new regions present new challenges for the effective control of RVFV epidemics. Changing climatic conditions can potentially impact daily environmental temperatures, which can influence mosquito densities, feeding behavior, fecundity, and the transmission dynamics of vector-borne diseases (VBDs). Mosquitoes tend to show optimal seasonal activity within ranges of specific meteorological parameters that increase their chances of survival [31]. It is important to note that the association between vector competence and temperature changes, presumably, may have a temperature limit, beyond which (temperature extremes), the influence of temperature may not necessarily be a determining factor or response to RVFV transmission. For instance, it is thought that the transmission of RVFV by vector species within their “normal temperature” range of activities may not be the only confounding risk to RVFV spatial reach in future epidemics and may include influences from diurnal average temperatures, where mosquitoes tend to experience daily fluctuating conditions [31,33]. Firstly, temperature affects the viability of eggs and when they hatch; different mosquito species require different temperatures for optimal hatching times [84]. Carrington et al. [85] demonstrated that temperature changes affect life-history characteristic estimations in both immature and adult *Ae. aegypti*, with the extent of biological changes influenced by the degree of daily temperature ranges (DTR). Diurnal temperature regimens (spanning 24 °C to 35 °C) on larval ambient temperature impact the developmental rates of *Ae. aegypti*, with significantly more females emerging than males [86]. However, higher temperatures have resulted in a decreased proportion of females emerging in the *Anopheles gambiae* species [87]. These findings demonstrate how temperature variations, influenced by inherent genetic heterogeneity, potentially induce phenotypic variations (plasticity) in mosquito species and their potential to influence RVFV spread [46]. Despite significant attempts at determining how thermal changes alter mosquito susceptibility and the transmission dynamics of RVF, establishing diurnal temperature conditions in the laboratory has proven a challenge to these studies. In spite of this, the very few studies that have closely mimicked these conditions have demonstrated that the RVFV vector, *Ae. albopictus*, raised under a DTR of between 23–31 °C recorded increased rates of colony survival, oviposition, and egg hatching, albeit with a low pupation rate, compared to mosquito colonies raised under constant high-temperature conditions (31 °C). These findings marked a considerable difference in the phenotypic characteristics of *Ae. albopictus* relative to those that were kept at a constant temperature [88]. Similarly, virus thermodynamic modeling predicted that when variations occur at mean temperatures over 18 °C, *Ae. aegypti* competence for flaviviruses was projected to decrease, whereas transmission was likely to rise at mean temperatures below 18 °C [89]. Specifically, thermodynamic simulations demonstrated that small temperature changes close to an average temperature of 29 °C during DTRs boost DENV transmission potential in tropical regions, whereas a higher DTR reduces these effects. DENV transmission has been shown to increase with increasing DTR in cold temperate or searing climates [85,89,90]. Wherefore, the potential effects of DTR on the transmission probability of DENV are shown to be greater than on infection probability owing to longer mosquito survival and likely infection under moderate temperature changes [89]. A predictive study using generalized linear mixed models indicated the likelihood of RVFV hotspots where *Ae. vexans* is a vector that was adversely impacted by the relative humidity and maximum and minimum temperatures, compared to *Cx. poicilipes*, which were positively correlated with a drop in the minimum temperature [91].

#### 3.1.2. Empirical Findings

Some evolutionary studies have pointed to population heterogeneity being a driving force of temperature-dependent adaptations generally, impacting RVFV transmission (as well as short vector generation times between hatching eggs and emerging adults). These increases in densities between mosquito species coinciding with epidemic peak periods significantly influenced disease dynamics [92]. Furthermore, the prevalence of particular species and the outbreak of RVF epizootic and epidemics at a given time may contribute to the intricacies of determining any linkages that exist between the mosquito species responsible for transmission [32,93]. Observations of different transmission efficiencies between mosquito species have been implicated to be a result of vector sensitivity to changes in daily precipitation and temperatures, particularly in the generation of eggs and rate of hatching, the pace of larval metamorphisms (instar development) to adults, life span, and biting frequencies [85]. *Aedes* mosquitoes and mosquitoes of the *Eretmapodites* genus were first suspected and confirmed as vectors primarily responsible for perpetuating RVFV transmission in Kenya [94,95]. Cohort vector surveillance studies led to the detection of RVFV in high numbers of *Culex*, *Anopheles,* and *Mansonia* species implicated in disease transmission, following periods of heavy rainfall, and exposure to infectious bites which exacerbated RVFV spread [29,30], indicating the pervasive potential for transmission of the virus by different vector genera. Also, a previous study reported a change in the RVFV vector-dominant species toward *Mansonia uniformis* and *Cx. poicilipes* during a second major RVF epidemic in 1998–1999 in Mauritania (research performed on both sides of the Senegal River Basin) [96]. This scenario highlights the susceptibility of mosquitoes that are native to RVF endemic and nonendemic regions and their capacity to serve as vector species during a disease outbreak. Mosquito development can be sped up in warmer climates [97] as well as influence year-round mosquito reproduction, thus extending the risk of transmission of RVFV [29,97,98,99]. For example, during the El Niño Southern Oscillation (ENSO), weather patterns were determined to be a major contributor to changes in the vector population preceding RVFV epizootics [98]. The succession of vector emergence from dambos and subsequent involvement in transmission is also tightly linked to precipitation [100]. Previous entomological survey studies in Senegal found that some RVFV vectors, *Cx. tritaeniorhynchus* (*p* = 0.04), *Ae. vexans arabiensis* (*p* < 0.001), and *Ma. uniformis* (*p* = 0.01), showed an increased prevalence as temperatures increased, while *Cx. poicilipes* prevalence decreased (*p* = 0.003) [101].

That temperature changes can significantly impact VBD transmission has been shown by vector competence. Daily environmental temperature changes have impacted DENV vector competence, with a high DTR (mean ~20 °C) found to lower DENV infection rates in the midgut of *Ae. aegypti* [89,102]. Additionally, empirical results regarding the effect of environmental temperature changes on vector competence for RVFV in mosquitoes found that *Cx. pipiens* adults held at higher temperatures (26 °C or 33 °C) were more susceptible to RVFV infection compared to a lower holding temperature (13 °C) [103]. Also, *Ae. fowleri* has been shown to have no discernible change in RVFV infectivity at comparable temperatures of 17 °C, 28 °C, and DTR 17–28 °C) [104]. The effects of temperature on vector competence variability across different mosquito genera and species, and the risk of transmission over similarly large territories would be better determined by extensively investigating the vector competence—or lack thereof—for all of these vectors, especially in endemic regions from an empirical perspective.

### 3.2. Extrinsic Incubation Period and Holding Temperature Conditions Are Important in RVFV Transmission

Fluctuating environmental temperature has been shown to affect *Ae. aegypti* and *Ae. albopictus* vector competence, inducing a strong purifying selection of ZIKV in infected *Aedes* species. Conversely, constant medium temperature (28–32 °C) was determined to have a positive selection with the same vector competence effect (overall low vector competence) at the set temperature extremes, i.e., the lowest temperature point at 16 °C and the highest temperature point of 38 °C) on ZIKV transmission [105,106]. However, in other arboviruses such as RVFV, West Nile Virus (WNV), and St. Louis Encephalitis virus (SLEV), several observations have been made regarding vector competence and extrinsic incubation period (EIP); the time between a mosquito imbibing on the infected bloodmeal to when the virus is then transmitted by bite [107,108,109,110,111,112]. For example, the EIP in *Cx. pipiens* and *Ae. taeniorhynchus’s* capacity to transmit RVFV was impacted significantly by differences in environmental temperature [103]. When mosquitoes were maintained at high temperatures, virus dissemination and transmission were established early in both species relative to lower temperatures. In *Ae. taeniorhynchus* mosquitoes maintained at temperatures of 13, 26, and 33 °C had comparatively stable RVFV infection rates of 55%, 56%, and 59%. Alternatively, holding conditions of 26 °C (75%) or 33 °C (91%) resulted in a much higher RVFV infection rate of *Cx pipiens*. Generally, at higher holding temperatures, RVFV infection rates of *Cx. pipiens* were significantly higher compared to *Ae. taeniorhynchus*. In terms of the effects of holding temperature on the EIP, *Cx. pipiens* were observed to be permissive to RVFV transmission in a shorter period, one day post-inoculation, following incubation at 26 or 33 °C. However, it took three days post-inoculation for observed transmission in the *Ae. taeniorhynchus* colony [103].

Additionally, studies on the effects of EIP on mosquito vectors’ developmental stages showed that a decreased holding temperature (at 15 °C) resulted in a significant increase in the survival span of the adult stages of both species. This phenomenon was theorized to further increase vector presence and chances of its exposure to hosts during biting activities in an ongoing outbreak [113,114]. These observations further call into question the potential impact of temperature conditions on the evolution and transmission dynamics of RVFV, as temperature fluctuations can alter the virus and the mosquito genetics [115], which tend to influence mosquito survival, susceptibility to infection, and virus persistence.

### 3.3. Impact of Temperature on RVFV Replication and Fitness of Emerging Genotypes

The genomic organization of RVFV, like many other viruses within the *Bunyavirales* order, has tripartite negative-sense, single-stranded RNA genome segments. These segments are the small (S), medium (M), and large (L) segments, making up a combined length of approximately 11.9 kb [116]. Phylogenetic and sequence analyses of the RVFV genome revealed a substantially conserved overall gene structure [117,118,119], with the diversity in the genomic structure found to be approximately 4% and 1% at the nucleotide and amino acid levels, respectively [29,120]. However, the low evolutionary diversity within the RVFV genome can be attributed to the absence or below detectable limits of definitive serotypes depending on the hosts. Furthermore, intrinsic features of the virus replication mechanism, virus–host interactions, and other environmental stressors can induce bottlenecks contributing to virus titers and detection [121]. Also, assessments performed from a geographical, evolutionary, and molecular point have shown that some viruses have evolved as a result of slow environmental changes over the years [119]. Coupled with the poor efficiency of the RNA-dependent RNA polymerase (RdRp) of many RNA viruses, a dynamic mutant cloud of closely related genotypes can be introduced, termed quasispecies, during the genome replication and reassortment of the virus [15,122,123]. The RdRp of RNA viruses has been shown to be temperature-sensitive, which contributes to the high rate of mutation occurrence during replication [124]. RNA viruses experience high mutation rates due to a relatively unstable RNA as the genomic store of information relative to DNA. However, for RVFV, this typically has resulted in low genetic variations in the RVFV genome (~1–5%), partly due to the challenge of quantitatively identifying intragenic recombinant events. Additionally, viral strains must at least simultaneously co-infect a single cell for natural reassortment to occur among genome segments [125,126]. Therefore, we cannot ignore the fact that with the length of time it takes for holometabolous hosts, such as mosquitoes (life cycle involving four stages), to mature and the time needed for RVFV to propagate, generating infectious titers under temperature-sensitive conditions might coincide favorably with influencing virus dissemination and transmission. Accordingly, temperature changes have been found to intricately shape the structure and function of nucleic acids, proteins, and the lipid organization of these viruses [127]. Furthermore, temperature has been observed to have a direct influence on the genome of RVFV during reassortment, as illustrated in some studies conducted on *Bunyavirales* serogroups acquired using normal temperature manipulation techniques [128]. Characterization of reassortant viruses produced between the vaccine strain and a wild RVFV strain from Senegal allowed researchers to determine the changes causing MP12 virus attenuation [129].

Temperature-sensitive mutations from these experiments were discovered in the L segment of the MP12 strain after testing for virulence traits of the reassortant viruses in mice. This study showed that each genome segment included at least one mutation that could independently limit the virus’s pathogenicity, confirming traits subject to polygenic regulation. In support of these findings, the authors further demonstrated that a reversion to virulence was improbable and that instead, attenuated genotypes were anticipated to emerge via genetic reassortment with wild-type viruses during a vaccination campaign in endemic regions [129]. Additional co-infection studies showed that reassortment and selection for temperature-mutants originating from distinct RNA regions of viruses occur when “defective” viruses (reassortants) and the standard viral segments are cultured together at a restrictive temperature [128]. For instance, temperature-sensitive MP12 reassortants were readily recovered from multiply infected Vero cells in the presence of 5-fluorouracil [129]. Mosquitoes as ectothermic vectors can adapt to the surrounding temperature, which raises further questions about the mechanism of selection, host interactions, and identification of gene markers that influence RVFV transmission and disease pathogenesis. Therefore, it was hypothesized that any massive mutant repositories emerging could potentially adapt to new surroundings of their host after bypassing bottlenecks such as the host immunity, antiviral treatments, or the selective pressures of temperature [130,131]. This has been demonstrated with the re-emergence of RVFV lineage H in Senegal and a potential to influence the disease transmission in the West Africa region [132].

#### The Implications of Environmental Temperature on Intraspecies Competition—Assessing Temperature Sensitivity in Virulent and Attenuated Strains of RVFV

While vector–virus interactions are influenced by host immune responses and the virulence of viruses [133], the ability to traverse barriers to infection can lead to genetically selected populations [134]. This further raises questions about the mechanism of selection, host interactions, and fitness of potential viral quasispecies within vector species. Additionally, the concerted effects of temperature and mutation-driven alterations among virus genotypes may impact virus survival in the mosquito and subsequent transmission.

Mutation-driven diversification can introduce a wide range of genetic combinations via the exchange of viral segments, allowing for adaptations that confer a biological benefit in viruses. The evolution of most DNA and RNA viruses is attributed to mutation, recombination, and segment reassortment [135]. Although the coordinated action of recombination or reassortment with mutation can occur at different times among segmented RNA virus families, including the *Orthomyxoviridae* (influenza virus) and *Bunyavirus* (RVFV) families [136], mutation remains a ubiquitous genetic alteration that underpins many adaptive responses and important biological changes in these virus families over time. It also serves as a necessary condition for recombination and reassortment to have a biological effect [135].

According to analysis of the available genomes, reassortment among bunyavirus serogroups may be more common than previously thought [137,138], with the Ngari virus exemplified as a reassortant virus from a recombination of the M segment of Batai virus (BATV) and the L and S segments of BUNV [139] (Appendix A). Geographic coincidence with BATV, one of the most prevalent orthobunyaviruses, is implicated in the conception of ecological reassortment. The query remains whether there is the possibility that some segments of NRIV isolates could match more closely than those of BATV isolates from more widespread geographic locations [139]. Generally, bunyaviruses have shown high titers of defective genotypes with repeated passaging over time. These defective mutants subsequently interfered with the normal replication of the non-defective (wild) virus strains [128]. Genetic analysis has shown that several mutations on both the S and M segments of the MP-12 genome conferred attenuation. However, other studies investigating the roles played by the Gn and Gc envelope proteins as well as the L protein in viral virulence were contrasted, where Morrill et al. [122] showed that a single nucleotide substitution within the wild-type RVFV Gn gene significantly hindered the virus’s pathogenicity in mice but later resulted in a rapid reversion or accumulation of virulent RVFV in mice from the attenuated strains [122]. Although the conclusion about reassortment between the vaccine strain MP12 and wild-type RVFV is that this is not a concern, a theoretical conception was that the mechanism of reassorting between closely related genotypes might hinder vaccination programs should there be reassortment between the MP12 attenuated ts strain and the wild-type RVFV strain [140]. Studies assessing the replication kinetics between RVFV attenuated strains (ZH548-M 12 and T1) and temperature sensitivities in hamsters found that the attenuated strains not only demonstrated a transient replication capacity due to their low pathogenicity but that the replication of these strains (ZH548-M12 and T1) was also impacted by the hamsters’ holding temperatures [141]. Interestingly, temperature sensitivity in the T1 infected group enhanced viremia significantly at 35 °C relative to hamsters held at a 37 °C temperature [141]. Also, in contrast, ZH501 and T46, both virulent strains of RVFV, generated high titers in the hamsters, displaying no significant sensitivity to temperature variations. These observations were confirmed in cell culture investigations, with both the ts strains successfully replicating in Vero cells from further away, producing an apparent cytopathic effect (CPE) at 35 °C [141]. Despite the conventional concerns of reversion to virulence of attenuated vaccine strains during vaccination campaigns in endemic regions, there are no concerns of genetic reassortment between vaccine strains with wild-type viruses [129,142]. There have also been studies showing the reversion of mutations in temperature-sensitive regions of MP12 following serial passaging in cell cultures [142]. The divergence in the genotypes in response to temperature changes hinders broader conclusions and emphasizes the need for further investigations on the virus’s evolutionary characteristics.

### 3.4. The Effect of Mosquito Eco-Immune Responses to Thermal Changes on RVFV Transmission

According to Bowden et al. [143], eco-immunology assesses the interplay between the host evolution, ecology, and physiology of a pathogen’s transmission under subversive environmental conditions [143]. To sustain fitness and adapt their phenotypes to environmental stressors like temperature, mosquitoes frequently deploy gene regulatory mechanisms that maintain metabolic homeostasis [144,145]. As temperature has the potential to alter the conformational structure and function of immune cells and enzymes, different mosquito species could present different thermal capacities contingent on their climate-adaptability to ensure the optimal or normal activity of vectors’ immune responses [146]. Therefore, in response to viral infections under varying thermal conditions, various components of the immune regulatory processes in mosquitoes may be modified to maintain metabolic homeostasis. Although field-based studies can be an effective assessment of eco-immunology, where the vertebrate host or invertebrate vectors are subjected to a wide range of ecological stimuli in their natural habitat, controlled laboratory studies can provide valuable insights into the effects of eco-immunity on virus transmission dynamics. In such settings, Muturi et al. [147] demonstrated that the fourth instar larvae of *Ae. aegypti* exhibited differential expression of immune-specific and detoxifying genes at 32 °C [147], increasing their survival and number at adult stages.

Over 30 different species belonging to the *Aedes* and *Culex* genera are known to transmit RVFV. Therefore, given the differences in the primary environmental determinants and the resultant distributions of species across their genera, it is hypothesized that each species’ response to climate change might also be distinct [148,149], and this phenomenon could impact RVFV transmission rates. For example, evidence of this functionality has been demonstrated where heat shock proteins (Hsp) expressed by mosquitoes act as molecular chaperones responsible for restoring a normal balance of metabolic functions within the vector after a blood meal [150]. Although several Hsps have been marked as significantly involved in thermoregulation pathways in response to high-temperature exposure in mosquitoes and other insects, Hsp70 happens to be extensively investigated [150,151,152]. In one study, Hsp70 expression in *Ae. aegypti* mosquitoes increased significantly in response to increasing thermal changes and other stressors [150]. Additionally, these proteins were distinctively or highly expressed within the midgut region after a warm blood meal. These assertions were supported by suppressed expression of Hsp70 via RNA interference, which caused decreased digestion of blood proteins after imbibing on a warm blood meal [150]. Similarly, *Cx. pipiens* and *Anopheles gambiae* species showed decreased egg numbers following RNA interference (RNAi) with Hsp70 expression [150]. These findings highlight an important consideration of the effect that thermal stressors induced by imbibing on a blood meal may have on the replication or transmission mechanisms of RVFV. Additionally, in mosquitoes, the RNAi pathways represent the most prominent antiviral response [153], with studies showing that the rearing of mosquitoes at low rearing temperatures (18 °C) disrupts priming of the RNAi pathways [154]. It has been observed that inhibiting the RNAi machinery in mosquito cells increases the replication of both RVFV and other bunyaviruses. This was demonstrated in vitro when RVFV was found to trigger the production of virus-derived piRNAs in *Cx. quinquefasciatus* mosquitoes [155]. Furthermore, the RVFV M segment, which codes for P78, and other ts proteins, was shown to be involved in subjugating the midgut physical barrier, aiding in the dissemination of RVFV in some mosquito species such as *Ae. aegypti* [156]. Adelman et al. [154] found that rearing *Ae. aegypti* colonies at lower temperatures resulted in impairment of the RNAi pathway, which increased susceptibility to CHIKV and yellow fever virus (YFV) [154]. Exogenous siRNA and piRNA, two antiviral RNAi pathways, have been demonstrated to be triggered by RVFV infection in vector species, including two of the most important mosquito genera for RVFV transmission. The significance of RNAi pathways in viral infection in mosquito species is shown by their ubiquitous function, and processes across these vector–virus systems may be conserved.

Heat shock proteins, especially those belonging to the Hsp90 and Hsp70 families, have significant functions during RVFV infection, according to inhibition experiments employing well-characterized small-molecule Hsp inhibitors [157]. Hsp70 kDa protein 5 has been demonstrated to be down-regulated by some siRNA constructs during RVFV replication, which reduces RVFV infection [157]. On the other hand, the downregulation of hsp70 kDa protein 8 demonstrated the opposite impact through increased viral infection [157]. Additionally, these studies demonstrated that, despite a significant decrease in total viral levels, the virions’ functional ability remained unaffected by strong Hsp inhibition. This suggests that, although Hsps are crucial for viral production, they may not always have an impact on infectivity within the virions [157]. This creates more opportunities to investigate the potential impact of eco-immune responses on the spread of RVFV in these mosquitoes.

### 3.5. Gut Symbionts, Co-Infections, and Temperature Interdependence—“Proponent or Detractor” of RVFV Transmission Control

Temperature can significantly impact the larval ecosystem of mosquito vectors, fundamentally shaping the microbial profiles of the colony [158,159,160,161]. The importance of these microbial populations cannot be overemphasized as they play a significant role in vector nutrition, immune response, competence, and development [162,163]. High densities of diverse microbial communities such as protozoans, fungi, bacteria, and insect-specific viruses (ISVs) tend to colonize the midgut of mosquitoes [164,165]. For example, in Kenya, microclimatic differences in temperature and humidity were determined to alter the microbial communities of wild-caught *Aedes* species, thus impacting the spatio-temporal transmission dynamics of RVFV [166]. Other studies reported that a drop in temperature elicited a compositional change in the midgut microbiota of *Ae. albopictus* [167], the mosquito species implicated in RVFV transmission in the study area (Kenya).

The mosquito midgut microbiota have been extensively studied in recent years, which could eventually lead to the manipulation of some of these bacterial communities as mitigation strategies for VBD transmission [168]. This is evident in contributions from Novakova et al. [169], who showed that warmer temperatures decreased the concentration of *Wolbachia* in *Cx. pipiens/restuans* mosquitoes, resulting in high vector susceptibility to WNV infection and transmission [169]. Similarly, mosquitoes infected with *Wolbachia* had a considerably greater rate of WNV infection than uninfected controls in *Cx. tarsalis* colony mosquitoes [170]. This was in contrast with *Cx. tarsalis* infected with RVFV, such that high densities of *Wolbachia* induced no significant change in RVFV infection but marginally reduced RVFV densities in *Cx. tarsalis* in a density-correlated manner [171].

While these reports highlight the important role of bacteria, other studies suggest that vector-driven genetics altered local adaptations between hosts and parasite populations as well, driving the patterns of disease outbreaks and the evolutionary conveyance of DENV dissemination in mosquitoes [172]. This suggests that in addition to bacteria, protozoan symbionts can also impact virus replication and subsequent transmission. Additionally, co-infection studies have demonstrated that ISVs can influence the transmission kinetics of WNV, ZIKV, and DENV [173,174]. The interdependence of ISVs and other microbiomes to further enhance virus transmission potential by mosquitoes may occasionally produce conflicting or unclear outcomes. Until now, there has been a single reported study of the role of ISV, *Cx. flavivirus* (*CxFV*), in RVFV transmission [175], where proponents of the study argued that *CxFV* did not have any effect on the vector competence of *Cx. pipiens* in transmitting RVFV from experimental demonstrations [175]. Additionally, although RVFV infection had previously been determined to have no significant change on the immune effector gene expression levels in mosquitoes upon initial exposure to RVFV, immunological responses to the virus were subsequently activated by prior bacteria priming, limiting virus proliferation [176]. Intrinsically, these findings form the bulk of current knowledge about ISVs and RVFV co-infection studies, highlighting a need for broader investigations. Conversely, it remains to be determined the extent to which temperature effects shapes the midgut microbiota of mosquito vector and impacts RVFV replication and transmission.

## 4. Knowledge Gaps in RVFV Vector Competence Studies and the Influence of Temperature on Transmission

The expansion in mosquito populations and the increasing emergence of RVFV outbreaks are being linked to changes in environmental temperatures. Some studies have shown that temperature-dependent adaptations of arboviruses can be facilitated by distinct mutations and that the short generation times of vectors aid in adaptation to new environments. Thus, the potential risks for the global spread of climate-driven VBDs such as RVFV increase due to microclimatic changes. Given that mosquitoes are holometabolous (undergo complete metamorphosis), a balance between the mosquito’s antiviral immune response and the virus’s persistence can, in part, be affected by temperature changes, as discussed throughout the text. Additionally, recent studies call into question the conventional theory that only the adaptive immunity of the mosquito can foster immunological memory, as proven in their lack of T and B lymphocytes, yet the ability to occasionally present strong resistance to reinfection via their innate immune responses [177]. These findings present more questions regarding what mechanisms these bottlenecks play in the mosquitoes’ susceptibility or resistance to infection and dissemination of the virus. A precise understanding of these events, especially RVFV’s natural transmission mechanism during interepizootic times, may explain current theories related to the virus’s long-term survival in mosquito species and transovarially infected eggs. Furthermore, exposure to the virus and the extent to which the immune regulatory processes of these potential vectors impact the progression of the virus during transmission cycles at the various developmental stages of the vectors are not fully characterized. Finally, the effect of larval habitat temperature on the microbial populations of vectors cannot be overemphasized as this among other factors presents a risk for disease spread. Therefore, to bridge these gaps in knowledge, it is important to initiate broader investigations into the role of temperature changes in RVFV evolution, mosquitoes’ susceptibility, and the persistence of the disease.

## 5. Conclusions

The frequency of climate-driven epidemics of RVFV, the wide range of vector susceptibility, and the concerns for global expansion of the disease require more effective efforts to address these threats. Some ecological modeling studies have presented an opportunity for forecasting RVFV outbreaks, with empirical studies having validated some vector competence parameters in relation to temperature fluctuations. Significant advancements in technologies and improvements in laboratory setups have further permitted the establishment of field diurnal conditions for vector-borne disease investigations. Also, these studies have provided important information on the range of competencies of mosquitoes, from almost non-competent to moderately competent vectors of RVFV transmission. However, disparities in vector competence remain due to the increasing changes in microclimatic temperatures globally, making it more difficult to draw broad conclusions about virus evolution, vector susceptibility, and RVFV transmission, especially in temperate regions. This is further compounded by increases in mosquito heterogeneity, densities, biting rates, and survival rates—questions that remain unanswered and prompt the need for broader investigations. This review provides important updates to these studies and highlights some knowledge gaps for consideration in future studies.

## Figures and Tables

**Figure 1 viruses-17-00217-f001:**
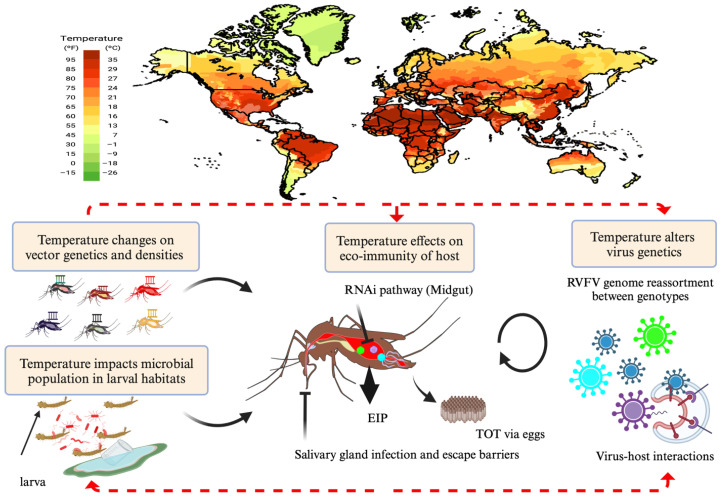
Overall, temperature can impact the natural habitats of mosquitoes (Section 2) and virus genetics (Section 3). In response to environmental stresses like temperature, mosquitoes exploit inherent gene regulatory systems to preserve fitness and modify their phenotypes (Section 3.1, Section 3.2, Section 3.3, Section 3.4 and Section 3.5) indicated by the red dotted arrows. The black arrows indicate various factors that contribute to mosquito vector capacity and RVF transmission. In effect, temperature can modify key elements of vector competence such as the microbiome (Section 3.5) shifting the balance in favor of increased or decreased virus transmission and persistence via modifications to transmission mechanisms including TOT (Section 2.1). EIP: extrinsic incubation period; TOT: transovarial transmission. This figure was created in BioRender. Lab, K. (2024).

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
