# Peer review of "The Temperature-Associated Effects of Rift Valley Fever Virus Infections in Mosquitoes and Climate-Driven Epidemics: A Review"

_viruses, 2025, doi:10.3390/v17020217_

Round 1
Reviewer 1 Report
Comments and Suggestions for Authors
The review "The temperature-associated effects of Rift Valley fever virus infections in mosquitoes and climate-driven epidemics: A review" considers the impact of temperature on various aspects of RVFV transmission. The link to climate change is weak but the concept is well worth investigating as environmental temperature affects both the biology of the mosquito vector and the effective replication and transmission of virus. However, the review is long and would benefit from reduction to those topics that are clearly defined, supported by evidence and relevance to Rift Valley fever virus. In places the English could be improved to convey the message clearly. The following comments are for consideration by the authors:
Line 16, change to 'environmental temperatures' and delete 'subsequently'.
Line 20, change to '..vector competence, and virus persistence during interepidemic years. This intricate...'
Line 43, change to 'United States Department of Agriculture'
Line 50, change to '... vaccine strains, Smithburn and Clone-13..'
Line 63, suggest replacing 'questing' with 'feeding' as it is the act of feeding that enables transmission, not questing alone.
Line 80, delete 'the' ahead of 'disease spread'
Line 85, change to '..that potentially improves...'
Line 109, delete 'the' ahead of 'disease'
Line 122, suggest replacing 'much' with 'large' and below, replace 'also well suited' to 'adapted'
Line 185, deleted 'a' ahead of 'transovarial transmission'
Line 230. It is not clear what the authors are suggesting in the sentence 'These findings demonstrate how temperature changes alter genetic heterogeneity of mosquito species and its potential to influence RVFV spread [46]'. Are they suggesting that temperature alone will alter the genetics of mosquitoes (which seems unlikely) or that temperature will induce phenotypic changes (plasticity) underpinned by existing genetic variation (heterogeneity)? It is not clear from the text what they are trying to imply.
Line 265, change to 'precipitation'
Line 269. The sentence 'Later studies led to the discovery of heighten (?) populations of Culex, Anopheles and Monsonia species inculpated(?) in the disease transmission, following periods of heavy rainfall, and exposure to infectious bites which exacerbated RVFV spread [29,30].' Can the authors explain what this sentence is describing.
Line 277, revise to '.. nonendemic regions and their capacity to...'
Lines 287 - 295. These lines discuss Zika virus so unclear what relevance they have to RVFV, suggest reduce to a single sentence statement or delete entirely.
Line 344. Is the statement that '..the low evolutionary diversity within RVFV genome can be attributed to the absence of below detectable limits of definitive serotypes depending the host.' actually true. There is only one serotype, this is reflected in limited variation in the structural proteins but low levels of diversity across the genome cannot be attributed to serological drivers alone.
Line 350. What is the relationship between point mutations, quasispecies and re-assortment? The authors appear to be conflating different processes. How does the virus polymerase influence re-assortment, and how is this affected by temperature. This section is not well explained and should be revised.
Line 389. Should the subtitle be in italic.
Line 397, the sentence here does not make sense and appears similar to the point on line 350. Why is mutation-driven diversification required for re-assortment?
Line 409. Can the authors explain why Ngari virus indicates phenotypic plasticity (an ability of an organism to change in response to stimuli or environmental inputs) within this virus group?
Line 440. The section on 'Mosquito eco-immunity responses to thermal changes on RVFV transmission' is difficult to follow and in places does not make sense. On line 456, what does the statement 'spurring on their survival and furthering presence adult colonies.' mean?
Lines 460-470. What is the relationship between Hsp70 and Rift Valley fever? A link between the two is not made within the paragraph.
Line 475-477. What is the link between transovarial transmission and imbibing a 'warm blood meal'? In theory all blood meals will be the temperature of the host (and if mammalian or avian likely to be warmer than ambient). However, it is not clear how this might influence subsequent events that results in TOT. Suggest revise section 3.4 for clarity and direct relevance to RVFV.
Line 495. Section 3.5 was the least convincing and this reviewer would remove it. Many of the statements are vague, as is are the impacts of the microbiome on virus transmission. For example, reference 159 observed a difference in microbiome composition was related to difference in location. That is entirely plausible, but then speculated that this may influence RVFV transmission but provided no evidence for a causal link between the two. Paragraph 508-517 is related to West Nile virus. The remaining paragraph again discusses other viruses and speculates about RVFV.
Line 538, what is 'fronting' in the context of this heading.
Line 553, suggest replacing 'aversion (a strong dislike)' to 'resistance'
Line 578, suggest revise to '..and prompt the need for ...'
It is not clear what the purpose of the supplementary table is for other than augment the reference section. Suggest removal or revision and inclusion in the manuscript with an explanation.
Comments on the Quality of English Language
All comments provided above.
Reviewer 2 Report
Comments and Suggestions for Authors
In this review entitled « The temperature-associated effects of Rift Valley fever virus infections in mosquitoes and climate-driven epidemics: A review »,, F Azerigyik et al. explore the litterature to document the impact of temperature variation on Rift Valley fever virus / host and vectors interaction that may drive to epidemic.
To do that they both take exemple on other well know and studied arboviruses like dengue from the flavivirus genogroup and multiple couples of mosquitoes/virus. To note this is perfectly acceptable due to the high capacity of RVFV to be vectorized by numerous different mosquitoes species depending of the studied region and weather conditions.
The paper is well writen and if the question of weather temperature impact on arbovirus epidemic is largely studied and questionned, as it is in the following paper :
Widening geographic range of Rift Valley fever disease clusters associated with climate change in East Africa.
Situma S, Nyakarahuka L, Omondi E, Mureithi M, Mweu MM, Muturi M, Mwatondo A, Dawa J, Konongoi L, Khamadi S, Clancey E, Lofgren E, Osoro E, Ngere I, Breiman RF, Bakamutumaho B, Muruta A, Gachohi J, Oyola SO, Njenga MK, Singh D. BMJ Glob Health. 2024 Jun 10;9(6):e014737. doi: 10.1136/bmjgh-2023-014737.
The author propose a valuable new review on the subject.
I have only minor comment : the list of reference is large as expected following the subject but somme of this reference are hard to be retreived in the current form like : ref 5 or 6. IN general the reference to books or AWHO (OMSA ex OIE) are not easy to retrieve. The webpage should be coted or the doi.
I will also suggest a few more recent citation that may improve the meaning of this review.
A few recent reference deserve to be added like :
Vector Competence Assays for RVFV in Mosquitoes.
Birnberg L, Busquets N. Methods Mol Biol. 2025;2893:85-107. doi: 10.1007/978-1-0716-4338-9_8.
This technical paper that described a well defined method and process to allow comparable vector competence assays deserved to be cited. And the point is that in the large litterature cited by the authors there were also large variations in the exact definition of th vector competence. This should be carefully checked by the authors.
Seroprevalence of Rift Valley fever and associated risk factors in livestock of Afar Region, northeastern Ethiopia.
Megenas JA, Dadi ML, Mekonnen TK, Larrick JW, Kassa GM. Curr Res Parasitol Vector Borne Dis. 2024 Sep 16;6:100215. doi: 10.1016/j.crpvbd.2024.100215. eCollection 2024.
Comparative study of two Rift Valley fever virus field strains originating from Mauritania.
Chabert M, Lacôte S, Marianneau P, Confort MP, Aurine N, Pédarrieu A, Doumbia B, Ould Baba Ould Gueya M, Habiboullah H, Beyatt ABEM, Lo MM, Nichols J, Sreenu VB, da Silva Filipe A, Colle MA, Pain B, Cêtre-Sossah C, Arnaud F, Ratinier M. PLoS Negl Trop Dis. 2024 Dec 9;18(12):e0012728. doi: 10.1371/journal.pntd.0012728.
In this paper that deserved to be cited some clues about the molecular determinant of the RFVF virulence in the ongulates (RFVF isolated from camel versus goat here).
More importantly, the paper below evaluate the effectivement of recombination between different RVFV strain in vitro AND in vivo and thrully deserved to be cited and commented in the right place.
Rift Valley Fever Phlebovirus Reassortment Study in Sheep.
Balaraman V, Indran SV, Kim IJ, Trujillo JD, Meekins DA, Shivanna V, Zajac MD, Urbaniak K, Morozov I, Sunwoo SY, Faburay B, Osterrieder K, Gaudreault NN, Wilson WC, Richt JA. Viruses. 2024 May 30;16(6):880. doi: 10.3390/v16060880.
Tissue distribution and transmission of Rift Valley fever phlebovirus in European Culex pipiens and Aedes albopictus mosquitoes following intrathoracic inoculation.
Gardela J, Yautibug K, Talavera S, Vidal E, Sossah CC, Pagès N, Busquets N. J Gen Virol. 2024 Sep;105(9):002025. doi: 10.1099/jgv.0.002025.
Important paper about the vector competance of mosquitoes in temperate area with very large temperature variation (DTR). Ie the trouble is not limited to North America but is also rising concerns in Europe or as assessed in Madagascar or in Afar region or Ethipian with temperature variation associated to the altitude (heigh above sea ;-).
Mosquito dynamics and their drivers in peri-urban Antananarivo, Madagascar: insights from a longitudinal multi-host single-site survey.
Tantely ML, Guis H, Raharinirina MR, Ambinintsoa MF, Randriananjantenaina I, Velonirina HJ, Revillion C, Herbreteau V, Tran A, Girod R. Parasit Vectors. 2024 Sep 10;17(1):383. doi: 10.1186/s13071-024-06393-4.
An Entomological Investigation during a Recent Rift Valley Fever Epizootic/Epidemic Reveals New Aspects of the Vectorial Transmission of the Virus in Madagascar.
Tantely LM, Andriamandimby SF, Ambinintsoa MF, Raharinirina MR, Rafisandratantsoa JT, Ravalohery JP, Harimanana A, Ranoelison NN, Irinantenaina J, Ankasitrahana MF, Ranoaritiana DB, Randrianasolo L, Randremanana RV, Lacoste V, Dussart P, Girod R. Pathogens. 2024 Mar 16;13(3):258. doi: 10.3390/pathogens13030258.
And to be back to the role of environment in mosquito dynamics in these two paper with one focusing on periurban area where contact will be enhanced between vectors, ruminants seek animals and human ..
Here it is also surprizing and may add some clues about the review subject : even forest biome might be a RVFV target.
Evidence for circulation of Rift Valley fever virus in wildlife and domestic animals in a forest environment in Gabon, Central Africa.
Becquart P, Bohou Kombila L, Mebaley TN, Paupy C, Garcia D, Nesi N, Olive MM, Vanhomwegen J, Boundenga L, Mombo IM, Piro-Mégy C, Fritz M, Lenguiya LH, Ar Gouilh M, Leroy EM, N'Dilimabaka N, Cêtre-Sossah C, Maganga GD. PLoS Negl Trop Dis. 2024 Mar 1;18(3):e0011756. doi: 10.1371/journal.pntd.0011756. eCollection 2024 Mar.
The paper below discuss the role of recombination to explain RVFV virulence …
Genome characterization of Rift Valley fever virus isolated from cattle, goats and sheep during interepidemic periods in Kenya.
Onwongá AA, Oyola SO, Juma J, Konongoi S, Nyamota R, Mwangi R, Muli C, Dobi P, Bett BB, Ongus JR. BMC Vet Res. 2024 Aug 23;20(1):376. doi: 10.1186/s12917-024-04161-1.
The one below is focused on the replacement of a previous lineage of RVFV (G and C) by a new one previously shown in South-Africa highlighting the role of « impor-export » of the viral strains that may improve the chance of virulance gain in an epidemic.
Re-Emergence of Rift Valley Fever Virus Lineage H in Senegal in 2022: In Vitro Characterization and Impact on Its Global Emergence in West Africa.
Sene O, Sagne SN, Bob NS, Mhamadi M, Dieng I, Gaye A, Ba H, Dia M, Faye ET, Diop SM, Sall Y, Diop B, Ndiaye M, Loucoubar C, Simon-Lorière E, Sakuntabhai A, Faye O, Sall AA, Diallo D, Dia N, Faye O, Diagne MM, Fall M, Ndione MHD, Barry MA, Fall G. Viruses. 2024 Jun 25;16(7):1018. doi: 10.3390/v16071018.

Round 2
Reviewer 1 Report
Comments and Suggestions for Authors
No further comments